# Privacy Implications of Retrieval-Based Language Models

**Yangsibo Huang   Samyak Gupta   Zexuan Zhong   Kai Li   Danqi Chen**

Princeton University

yangsibo@princeton.edu   {samyakg,zzhong,li,danqic}@cs.princeton.edu

## Abstract

Retrieval-based language models (LMs) have demonstrated improved interpretability, factuality, and adaptability compared to their parametric counterparts by incorporating retrieved text from external datastores. While it is well known that parametric models are prone to leaking private data, it remains unclear how the addition of a retrieval datastore impacts model privacy. In this work, we present the first study of privacy risks in retrieval-based LMs, particularly $k$NN-LMs. Our goal is to explore the optimal design and training procedure in domains where privacy is of concern, aiming to strike a balance between utility and privacy. Crucially, we find that $k$NN-LMs are more susceptible to leaking private information from their private datastore than parametric models. We further explore mitigations of privacy risks: When privacy information is targeted and readily detected in the text, we find that a simple sanitization step would eliminate the risks while decoupling query and key encoders achieves an even better utility-privacy trade-off. Otherwise, we consider strategies of mixing public and private data in both datastore and encoder training. While these methods offer modest improvements, they leave considerable room for future work. Together, our findings provide insights for practitioners to better understand and mitigate privacy risks in retrieval-based LMs[1].

## 1 Introduction

Retrieval-based language models ([Khandelwal et al., 2020](); [Borgeaud et al., 2022](); [Izacard et al., 2022](); [Zhong et al., 2022](); [Min et al., 2023]()) generate text distributions by referencing both the parameters of the underlying language model and the information retrieved from a datastore of text. Specifically, the retrieval process involves accessing a pre-defined datastore to retrieve a set of tokens or

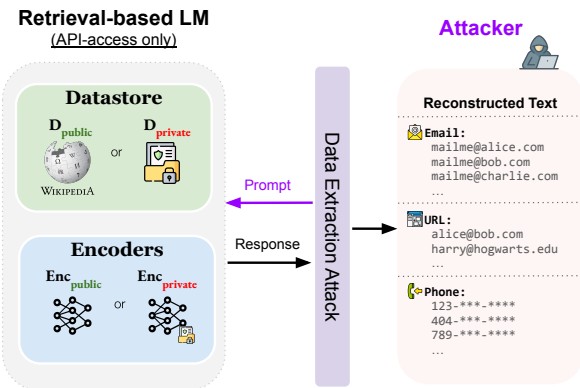

Figure 1: Retrieval-based language models (e.g., kNN-LMs) comprise encoders and the datastore as their key components. When tailoring a retrieval-based language model for a privacy-sensitive task, both components may utilize private data. However, a malicious user possessing API access to the model can exploit a data extraction attack in order to reconstruct sensitive information. This study explores the severity of such attacks and suggests novel strategies to mitigate them.

text passages that are most relevant to the prompt provided to the model. These retrieved results are then utilized as additional information when generating the model's response to the prompt. Retrieval-based language models offer promising prospects in terms of enhancing interpretability, factuality, and adaptability.

However, in privacy-sensitive applications, *utility usually comes at the cost of privacy leakage*. Recent work has shown that large language models are prone to memorizing ([Thakkar et al., 2021](); [Zhang et al., 2021]()) specific training datapoints, such as personally identifying or otherwise sensitive information. These sensitive datapoints can subsequently be extracted from a trained model through a variety of techniques ([Carlini et al., 2019](), [2021](); [Lehman et al., 2021]()), known as *data extraction attacks*. While the threat of training data memorization on model privacy has been studied for parametric language models, there is a lack

---

[1] Our code is available at https://github.com/Princeton-SysML/kNNLM_privacy.

of evidence regarding the privacy implications of retrieval-based language models, especially how the use of external datastore would impact privacy.

In this work, we present the first study of privacy risks in retrieval-based language models, with a focus on the nearest neighbor language models ($k$NN-LMs) (Khandelwal et al., 2020), which have been extensively studied in the literature (He et al., 2021; Zhong et al., 2022; Shi et al., 2022a; Xu et al., 2023)[2]. In particular, we are interested in understanding $k$NN-LMs' privacy risks in real-world scenarios where they are deployed via an API to the general public (Figure 1). We consider a scenario in which a model creator has a private, domain-specific dataset that improves model performance on domain-specific tasks, but may also contain sensitive information that should not be revealed; This data can be used to train encoders or stored in the datastore. In such a scenario, the model creator must find a balance between utilizing their private data to enhance model performance and protecting sensitive information.

We begin our investigation by examining a situation where the creator of the model only adds private data to the retrieval datastore during inference, as suggested by Borgeaud et al. (2022). Our findings indicate that while this approach enhances utility, it introduces an elevated privacy risk to the private data compared to parametric language models (Section 4), and adversarial users could violate the confidentiality of the datastore by recovering sensitive datapoints. Therefore, it is vital for the model creator to refrain from storing sensitive information in the datastore.

We further explore mitigation strategies for kNN-LMs in two different scenarios. The first is where private information is *targeted*, i.e., can be easily identified and removed (Section 5). We explore enhancing the privacy of $k$NN-LMs by eliminating privacy-sensitive text segments from both the datastore and the encoder's training process. This approach effectively eliminates the targeted privacy risks while resulting in minimal loss of utility. We then explore a finer level of control over private information by employing distinct encoders for keys (i.e., texts stored in the datastore) and queries (i.e.,

prompts to the language model). Through our experimental analysis, we demonstrate that this design approach offers increased flexibility in striking a balance between privacy and model performance.

The second is a more challenging scenario where the private information is *untargeted*, making it impractical to remove from the data (Section 6). To address this issue, we explore the possibility of constructing the datastore using public datapoints. We also consider training the encoders of the $k$NN-LM model using a combination of public and private datapoints to minimize the distribution differences between the public data stored in the datastore and the private data used during inference. Despite modest improvements from the methods we explored, the mitigation of untargeted attacks remains challenging and there is considerable room for future work. We hope our findings provide insights for practitioners to better understand and mitigate privacy risks in retrieval-based LMs.

## 2 Background

In this section, we first review the key components of $k$NN-LMs (Section 2.1). Then, we discuss the data extraction attacks for language models (Section 2.2). These aspects lay a foundation for the subsequent exploration and analysis of privacy risks related to $k$NN-LMs.

### 2.1 Nearest Neighbor Language Models

A $k$NN-LM (Khandelwal et al., 2020) augments the standard language model with a *datastore* from which it can retrieve tokens to improve performance. We use the term "query" to denote the prompt provided to a $k$NN-LM. This query is encoded by the encoder $\text{Enc}_Q$ as it is passed into the $k$NN-LM. The term "key" is used to denote the tokens in the datastore and it is encoded by the encoder $\text{Enc}_K$.

**Encoders** Given a vocabulary $\mathcal{V}$, the encoder $\text{Enc}_K$ or $\text{Enc}_Q$ performs the task of mapping a given key or query $c \in \mathcal{V}^*$ to a fixed-length vector representation. Typically, this encoding process is accomplished through a trained language model, where $\text{Enc}_K(c)$ or $\text{Enc}_Q(c)$ represents the vector hidden representation obtained from the output layer of the language model when provided with the input $c$. Although in the default $k$NN-LMs, $\text{Enc}_K$ and $\text{Enc}_Q$ are commonly identical functions, we explore different options in the work.

---

[2]Other retrieval-based language models such as RETRO (Borgeaud et al., 2022) and Atlas (Izacard et al., 2022) have significantly different architectures and the findings from our investigation may not necessarily apply to these models. Investigation of these models will be left as future work.

**Datastore** The datastore, $\{(\text{Enc}_K(c_i), w_i)\}$, is a key-value store generated by running the encoder $\text{Enc}_K(\cdot)$ over a corpus of text; Each key is the vector representation $\text{Enc}_K(c_i)$ for some context $c_i \in \mathcal{V}^*$, and each value $w_i \in \mathcal{V}$ is the ground-truth next word for the leftward context $c_i$. A search index is then constructed based on the key-value store to enable retrieval.

**Inference** At inference time, when predicting the next token for a query $x \in \mathcal{V}^*$, the model queries the datastore with encoded query $\text{Enc}_Q(x)$ to retrieve $x$'s $k$-nearest neighbors $\mathcal{N}_k$ according to a distance function $d(\cdot, \cdot)^3$. Then the model computes a softmax over the (negative) distances, which gives $p_{\text{kNN}}(y|x)$, a distribution over the next token, in proportional to:

$$\sum_{(c_i, w_i) \in \mathcal{N}_k} \mathbf{1}_{y=w_i} \exp\left(-\frac{d\left(\text{Enc}_K(c_i), \text{Enc}_Q(x)\right)}{t}\right),$$

where $t$ is a temperature parameter, and $k$ is a hyper-parameter that controls the number of retrieved neighbors. The prediction is then interpolated with $p_{\text{LM}}$, the prediction from the original LM: $p(y|x) = \lambda p_{\text{kNN}}(y|x) + (1 - \lambda)p_{\text{LM}}(y|x)$, where $\lambda$ is an interpolation coefficient.

## 2.2 Data Extraction Attacks

Prior work (Carlini et al., 2021) demonstrates that an attacker can extract private datapoints from the training set of a learned language model. The existence of such an attack poses a clear and alarming threat to the confidentiality of sensitive training data, potentially jeopardizing deployment in real-world scenarios (e.g., Gmail's autocomplete model (Chen et al., 2019), which is trained on private user emails). Our definition of privacy leakage adopts the standard definition of Carlini et al. (2021)). Specifically, a string $s$ is extractable from an $k$NN-LM $f_\theta$ if there exists a prefix $c$ such that: $s \leftarrow \arg\max_{s'} f_\theta(s'|c)$. Namely, the model generates $s$ as the most likely continuation when prompted with some prefix $c$.

The attack consists of two main steps: 1) generating candidate reconstructions by prompting the trained models, and 2) sorting the generated candidates based on a score that indicates the likelihood of being a memorized text. Further details about the attack can be found in Appendix A.

---

$^3 d(\cdot, \cdot)$ is usually the squared $\ell_2$ distance.

While previous research has successfully highlighted the risks associated with data extraction in parametric language models, there remains a notable gap in our understanding of the risks (and any potential benefits) pertaining to retrieval-based language models like $k$NN-LMs. This study aims to address this gap and provide insights into the subject matter.

## 3 Problem Setup

In this section, we formally describe our problem setup (Section 3.1) and privacy measurements (Section 3.2). We then detail our evaluation setup (Section 3.3).

### 3.1 Problem Definition

We consider a scenario where a service provider (e.g. a financial institution) aims to enhance its customer experience by developing a $k$NN-LM and deploying it as an API service. Note that the development of $k$NN-LMs intended solely for *personal use* (e.g., constructing a $k$NN-LM email autocompleter by combining a public LM with a private email datastore) falls outside the scope of our study because it does not involve any attack channels that could be exploited by potential attackers. We assume that the service provider possesses its own private data ($\mathcal{D}_{\text{private}}$) specific to its domain, in addition to publicly available data ($\mathcal{D}_{\text{public}}$).

We identify two key design choices which impact the quality and privacy of such a deployed service. First, the service provider chooses which data to be included in its datastore, and this may be public data ($\mathcal{D}_{\text{public}}$), private data ($\mathcal{D}_{\text{private}}$), or a mix of both. Second, they choose whether to use encoders that are pre-trained on publicly available data ($\text{Enc}_{\text{public}}$), or further finetuned on the private data ($\text{Enc}_{\text{private}}$). We posit that careful consideration of these design choices is needed to establish a balance between privacy preservation and utility.

The service provider in such a scenario is concerned with making a useful API, while keeping their private data hidden from malicious users or attackers. Hence, the service provider's objective is to attain a high level of utility (as measured by perplexity) on a held-out set of $\mathcal{D}_{\text{private}}$ while simultaneously minimizing the disclosure of private information. We quantify the metrics we consider for privacy in Section 3.2.

## 3.2 Privacy Measurements

We now describe how we evaluate the risk of data extraction attack within the scenario described earlier in Section 3.1.

**Threat model** We assume that the service provider deploys a $k$NN-LM with API access to $p(y|x)$. This API provides the attacker with the capability to compute perplexity, conduct text completion, and perform other relevant tasks. However, it's important to note that the attacker is restricted from accessing the internal parameters or the datastore of the deployed model.

Our study considers two types of privacy risks, each associated with a particular type of attack:

**Targeted attacks** We define targeted risk as a privacy risk that can be directly associated with a segment of text (e.g., personal identifiers such as addresses and telephone numbers.) and propose the targeted attack. The significance of a targeted attack becomes apparent when considering that targeted risks have been explicitly addressed in various privacy regulations (Centers for Medicare & Medicaid Services, 1996; California State Legislature , 2018). A targeted attacker's goal is to extract that certain segment of text. In our study, we focus on the extraction of Personal Identifiable Information (PII), including email addresses, telephone numbers, and URLs. To tailor the extraction attack to recover text segments such as PIIs rather than the entire training text, we customize the attack prompts based on the type of information to be extracted. Specifically, we gather common preceding context for telephone numbers, email addresses, and URLs, and use them as prompts. Appendix B provides example prompts we use in the attack. For evaluation, we measure how many private PIIs of each category have been successfully reconstructed by the attacker:

- We firstly detect all unique personal identifiers in the private dataset, denoted as $\{\rho_i\}_{i=1}^p$;
- We then sort the reconstruction candidates based on the membership metrics defined in Appendix A, and only keep the top-$n$ candidates $\{c_i\}_{i=1}^n$;
- Finally, we detect $\{\hat{\rho}_i\}_{i=1}^q$, the unique PIIs in the top-$n$ candidates, and then count $|\{\rho_i\}_{i=1}^p \cap \{\hat{\rho}_i\}_{i=1}^q|$, namely how many original PIIs have been successfully reconstructed by the attack. A larger number means higher leakage of private PIIs.

**Untargeted attacks** The untargeted attack is the case where the attacker aims to recover the entire training example, rather than a specific segment of text. Such attacks can potentially lead to the theft of valuable private training data. We adopt the attack proposed by Carlini et al. (2021) as the untargeted attack, which is described in detail in Appendix A.1. For evaluation, we measure the similarity between the reconstructed text and the original private text:

- We firstly sort the reconstruction candidates based on the membership metrics defined in Appendix A, and only keep the top-$n$ candidates $\{c_i\}_{i=1}^n$;
- For each candidate $c_i$, we then find the closest example in the private dataset $p_i$ and compute the ROUGE-L score (Lin, 2004) between $c_i$ and $p_i$[4]. If the score is higher than $0.5$, we mark the candidate as a good reconstruction. The ROUGE-L measures the longest common subsequence (LCS) between the attack result and the ground truth, thus representing the fidelity of the attacker results. The ROUGE-L score has been used to measure the reconstruction fidelity in previous work (Deng et al., 2021; Balunovic et al., 2022; Gupta et al., 2022).

Note that the attack's performance evaluation employs the private dataset following established reconstruction attack practices, the attack itself never utilizes this dataset.

## 3.3 Evaluation Setup

Our main evaluation uses the Enron Email dataset (Klimt and Yang, 2004) as the private dataset $\mathcal{D}_{\text{private}}$, which contains around 500,000 emails generated by employees of the Enron Corporation (see Table 1 for examples). We specifically chose this dataset due to its inclusion of PIIs, which enables us to evaluate targeted attacks effectively. We also incorporate the Medical Transcriptions dataset as an additional dataset for the evaluation of untargeted attacks, and further information regarding this dataset can be found in Appendix D. We use the WikiText-103 dataset (Merity et al., 2017) as $\mathcal{D}_{\text{public}}$.

We pre-process the Enron Email dataset by retaining only the email body (see Table 1 for exam-

---

[4]Note that Carlini et al. (2021) also focus on untargeted attacks but they adopt manual evaluation.

| | | Example #1: Request ID : 0000000000**** Request Create Date : //* 8:27:00 AM Requested For : ****.@enron.com Resource Name : Market Data Bloomberg Resource Type : Applications |

**Example #1**: Request ID : 0000000000**** Request Create Date : //* 8:27:00 AM Requested For : ****.@enron.com Resource Name : Market Data Bloomberg Resource Type : Applications

**Example #2**: You can reach me over the weekend and in the evening at either −**** or −****.

**Example #3**: This would give you a total loan of ****, total cost of **** for equity required of ****.

**Example #4**: Winter 2001 baseload traded as high as ** pounds a megawatt-hour and as low as **** pounds a megawatt-hour, before closing at **** pounds a megawatt-hour, **** pence lower than Friday.

**Example #5**: everything is correct except for password: ****.

Table 1: Examples from the Enron Email dataset. We've anonymized any identifiable information to protect against potential privacy breaches inherent in the dataset.

| MODEL | EVAL. PPL | TARGETED ATTACK | | | | UNTARGETED ATTACK |
|---|---|---|---|---|---|---|
| | | TOTAL | PHONE | EMAIL | URL | # GOOD RECON |
| (PARAMETRIC LM) $\mathrm{Enc_{public}}$ | 30.28 | 0 | 0 | 0 | 0 | 0 |
| (PARAMETRIC LM) $\mathrm{Enc_{private}}$ | 20.63 | 28 | 11 | 14 | 3 | 620 |
| ($k$NN-LM) $\mathrm{Enc_{public}}$ W/ $\mathcal{D}_{private}$ | 18.41 | 35 | 11 | 16 | 8 | 591 |
| ($k$NN-LM) $\mathrm{Enc_{private}}$ W/ $\mathcal{D}_{private}$ | 16.12 | 54 | 25 | 23 | 6 | 656 |

Table 2: Perplexity and data extraction risks for various model configurations on the Enron Email dataset. The configuration with the highest leakage is emphasized in red ; the configuration with the lowest leakage is highlighted in green . Privacy measurements are computed using the top 1000 candidates for the targeted attack and using the top 5000 candidates for the untargeted attack. "Good Recon" refers to reconstructions that achieve a ROUGE-L score greater than 0.5 when compared to the ground truth. Table 11 in Appendix D presents similar findings on the Medical Transcriptions dataset.

ple datapoints). We then use regular expressions to identify and extract three types of personal identifiers for the use of the targeted attack: telephone numbers, email addresses, and URLs. The statistics for these personal identifiers can be found in Appendix B. We use the GPT-2 base model (Radford et al., 2019) as $\mathrm{Enc_{public}}$, and finetune it on the Enron Email dataset as $\mathrm{Enc_{private}}$.

During inference, the model retrieves $k$ nearest neighbors according to the squared $\ell_2$ distance, normalizes their distribution over the next word with a softmax using a temperature value of 1, and then uses an interpolation factor of $\lambda$ to combine $p_{\mathrm{kNN}}$ and $p_{\mathrm{LM}}$. For each model configuration, we search $k$ and $\lambda$ on a held-out validation set for the best model perplexity, and then run the inference on the evaluation set (see Appendix B for details). Privacy measurements are computed using the top 1000 candidates for the targeted attack and using the top 5000 candidates for the untargeted attack.

## 4 Privacy-Utility of $k$NN-LMs with A Private Datastore

This section presents our investigation of whether the addition of private data to the retrieval datastore during inference is an effective method for achieving a good trade-off between privacy (mea-

sured by metrics defined in Section 3.2) and utility (measured by perplexity) in $k$NN-LMs.

We are particularly interested in three scenarios: utilizing only $\mathrm{Enc_{public}}$ (the publicly pretrained language model), utilizing only $\mathrm{Enc_{private}}$ (the model fine-tuned from $\mathrm{Enc_{public}}$ using private data), and utilizing $\mathrm{Enc_{public}}$ with $\mathcal{D}_{private}$ (the combination of the public model with the private datastore). As shown in Table 2, using $\mathrm{Enc_{public}}$ alone results in very poor utility performance but poses minimal risk of data extraction from the private domain, as it has not been exposed to private datapoints. Using $\mathrm{Enc_{private}}$ enhances utility (perplexity improves from 30.28 to 20.63) but increases the risk of data extraction.

When it comes to $k$NN-LMs, incorporating a private datastore ($\mathcal{D}_{private}$) with a public model ($\mathrm{Enc_{public}}$) yields even greater utility compared to relying solely on the fine-tuned model ($\mathrm{Enc_{private}}$). However, this utility improvement also comes at the expense of increased privacy leakage. These findings suggest that the privacy concern stemming from the private datastore outweighs that resulting from the privately fine-tuned model, indicating a lack of robust privacy protection in the design of $k$NN-LMs. Additionally, we note that the combination of $\mathrm{Enc_{private}}$ and $\mathcal{D}_{private}$ achieves the highest utility but also incurs the highest privacy cost.

| SANITIZATION APPROACH | SANITIZED ENCODER? Enc$_K$ | Enc$_Q$ | EVAL. PPL | TOTAL | PHONE | EMAIL | URL |
|---|---|---|---|---|---|---|---|
| NONE | ✗ | ✗ | 16.12 | 54 | 25 | 23 | 6 |
| REPLACED W/ < \|endoftext\| > | ✓ | ✓ | 16.83 | 0 | 0 | 0 | 0 |
|  | ✗ | ✓ | **16.24** | 0 | 0 | 0 | 0 |
|  | ✓ | ✗ | 16.32 | 16 | 6 | 6 | 4 |
| REPLACED W/ DUMMY PII | ✓ | ✓ | 16.51 | 0 | 0 | 0 | 0 |
|  | ✗ | ✓ | 16.29 | 0 | 0 | 0 | 0 |
|  | ✓ | ✗ | **16.16** | 22 | 5 | 15 | 2 |
| REPLACED W/ RANDOM PUBLIC PII | ✓ | ✓ | 16.38 | 0 | 0 | 0 | 0 |
|  | ✗ | ✓ | 16.29 | 1 | 1 | 0 | 0 |
|  | ✓ | ✗ | **16.20** | 23 | 8 | 13 | 2 |

Table 3: Perplexity and extraction risk of $k$NN-LMs with different combinations of key encoder (Enc$_K$) and query encoder (Enc$_Q$) on the Enron Email dataset, under different sanitization approaches. Privacy measurements are computed using the top 5000 candidates. For each sanitization approach, the configuration with the highest leakage is emphasized in red , and the configuration with the lowest leakage is highlighted in green ; the best utility under sanitization is highlighted in **boldface**. Sanitizing Enc$_Q$ is crucial to eliminating the targeted risk.

## 5 Mitigations Against Targeted Risks

Our previous findings indicate that the personalization of $k$NN-LMs with a private datastore is more susceptible to data extraction attacks compared to fine-tuning a parametric LM with private data. At the same time, leveraging private data offers substantial utility improvements. Is there a more effective way to leverage *private* data in order to achieve a better balance between privacy and utility in $k$NN-LMs? In this section we focus on addressing privacy leakage in the context of *targeted* attacks (see definition in Section 3.2), where the private information can be readily detected from text. We consider several approaches to tackle these challenges in Section 5.1 and Section 5.2, and present the results in Section 5.3. We also investigate the effect of hyper-parameters in Section 5.4.

### 5.1 Sanitization of Datastore and Encoders

As demonstrated in Section 4, the existence of private examples in the $k$NN-LMs' datastore increase the likelihood of privacy leakage since they are retrieved and aggregated in the final prediction. Therefore, our first consideration is to create a sanitized datastore by eliminating privacy-sensitive text segments. We note that this verbatim level definition of "privacy leakage" is general and widely adopted. Notably, regulations such as HIPAA (Centers for Medicare & Medicaid Services, 1996) and CCPA (California State Legislature , 2018) offer explicit definitions of privacy-sensitive data. Consequently, these regulatory frameworks can serve as the basis for establishing the verbatim-level definition of "privacy leakage". For example, HIPAA defines 18 identifiers that are considered personally identifiable information (PII), including names, addresses, phone numbers, etc.

We propose the following three options for sanitization:

- Replacement with < |endoftext| >: replace each privacy-sensitive phrase with the < |endoftext| > token;

- Replacement with dummy text: replace each privacy-sensitive phrase with a fixed dummy phrase based on its type. For instance, if telephone numbers are sensitive, they can be replaced with "123-456-789"; and

- Replacement with public data: replace each privacy-sensitive phrase with a randomly selected public phrase of a similar type. An example is to replace each phone number with a public phone number on the Web.

The encoders in a $k$NN-LM is another potential source of privacy leakage. While it is typically optimized on target domain data to enhance performance, fine-tuning directly on private data in privacy-sensitive tasks may result in privacy leaks (Table 2). Similarly, the encoder can be sanitized by fine-tuning the pre-trained encoder Enc$_{\text{public}}$ on a sanitized dataset that has had sensitive information removed.

## 5.2 Decoupling Key and Query Encoders

We propose using separate encoders for keys and queries in $k$NN-LMs, to allow for finer control over privacy preservation. For example, the encoder for queries can be the sanitized encoder, while the encoder for keys can be the non-sanitized one; This way, the query encoder can be more resistant to privacy leakage, while the keys encoder can provide better query results. While it is not a common practice in $k$NN-LMs, we view the separation of key and query encoders as a promising approach to reduce the discrepancy between the prompt and the datastore, and reduce privacy leakage.

The privacy risk of a $k$NN-LM can also be impacted by its hyper-parameters such as the number of neighbors $k$, and the interpolation coefficient $\lambda$. It is important to consider these hyper-parameters in the customization of the $k$NN-LMs to ensure that the privacy-utility trade-off is well managed.

## 5.3 Results of Sanitization and Decoupling Encoders

As demonstrated in Table 3, applying sanitization to both the encoder and the datastore effectively *eliminates* privacy risk, resulting in no personally identifiable information (PII) being extracted. Among the three methods, the strategy of replacing PII with random public information for sanitization yields the highest utility. It achieves a perplexity of 16.38, which is only marginally worse than the perplexity of 16.12 achieved by the non-sanitized private model.

Table 3 also demonstrates that utilizing separate encoders for keys and queries enhances the model's utility compared to using the same sanitized encoder for both. Specifically, we observe that when using the non-sanitized encoder for the query and the sanitized encoder for the key, privacy risks remain high due to the potential leakage from the $p_{\text{LM}}$. On the other hand, using the non-sanitized encoder for the key and the sanitized encoder for the query effectively eliminates privacy risk while still maintaining a high level of utility. This finding highlights the importance of sanitizing the query encoder in $k$NN-LMs.

## 5.4 Effect of Hyper-parameters

We finally analyze the impact of key hyper-parameters on utility and privacy risks in $k$NN-LMs, using $\mathcal{D}_{\text{private}}$ as datastore and $\text{Enc}_{\text{private}}$ for both $\text{Enc}_K$ and $\text{Enc}_Q$. First, we vary $\lambda$, the inter-

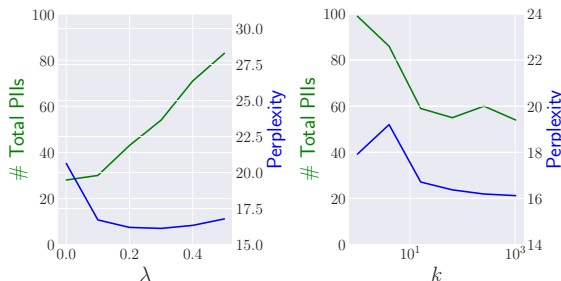

Figure 2: Effect of the interpolation coefficient ($\lambda$) and the number of nearest neighbors ($k$) on $k$NN-LMs' utility (measured by perplexity, the blue curve) and privacy risk (measured by the number of reconstructed PIIs in the targeted attack, the green curve). We use $\text{Enc}_{\text{private}}$ as encoders and $\mathcal{D}_{\text{private}}$ as the datastore.

polation coefficient, and observe that increasing $\lambda$ decreases perplexity but increases privacy risk (see Figure 2). This highlights the trade-off between accuracy and privacy, indicating that optimizing both factors simultaneously is challenging through $\lambda$ adjustment alone.

$k$ is the number of nearest neighbors in a $k$NN-LM. As shown in Figure 2, increasing the value of $k$ improves perplexity as it allows considering more nearest neighbors. We also notice that using a larger $k$ decreases privacy risk as the model becomes less influenced by a limited group of private nearest neighbors. Together, increasing $k$ seems to simultaneously enhance utility and reduce risk.

## 6 Mitigations Against Untargeted Risks

In this section, we explore potential methods to mitigate untargeted risks in $k$NN-LMs, which is a more challenging setting due to the opacity of the definition of privacy. It is important to note that the methods presented in this study are preliminary attempts, and fully addressing untargeted risks in $k$NN-LMs still remains a challenging task.

### 6.1 Methods

Considering that storing $\mathcal{D}_{\text{private}}$ in the datastore is the primary cause of data leakage (as discussed in Section 4), and the challenge of sanitizing private data in the face of untargeted risks, we propose the following approaches to leverage public data for mitigating these risks.

**Adding public data to datastore** The quality of the retrieved neighbors plays a crucial role in the performance and accuracy of $k$NN-LMs. Although it is uncommon to include public datapoints that are not specifically designed for the task or domain

| DATASTORE | | EVAL. PPL | # GOOD RECON |
|---|---|---|---|
| $N_{\text{pub}}$ | $N_{\text{priv}}$ | | |
| 0 | ALL | 16.12 | 656 |
| 0 | 0 | 20.63 | 620 ($\downarrow 5.5\%$) |
| ALL | 0 | 21.46 | 561 ($\downarrow 14.5\%$) |
| | 5,000 | 21.58 | 579 ($\downarrow 11.7\%$) |
| ALL | 10,000 | 21.23 | 595 ($\downarrow 9.3\%$) |
| | 50,000 | 19.16 | 617 ($\downarrow 5.9\%$) |
| | ALL | 19.02 | 632 ($\downarrow 3.7\%$) |

Table 4: Perplexity and data extraction risks for $k$NN-LMs with different numbers of public ($N_{\text{pub}}$) and private ($N_{\text{priv}}$) examples in the datastore. The encoder in use is the privately fine-tuned encoder $\text{Enc}_{\text{private}}$. Privacy measurements are computed using the top 5000 candidates. Risk reduction (%) compared to the first row is annotated. Table 12 in Appendix D presents similar findings on the Medical Transcriptions dataset.

| DATA STORE | $\text{Enc}_K$ | $\text{Enc}_Q$ | EVAL. PPL | # GOOD RECON |
|---|---|---|---|---|
| $\mathcal{D}_{\text{private}}$ | $\text{Enc}_{\text{private}}$ | $\text{Enc}_{\text{private}}$ | 16.12 | 656 |
| $\mathcal{D}_{\text{public}}$ | $\text{Enc}_{\text{private}}$ | $\text{Enc}_{\text{private}}$ | 21.46 | 561 ($\downarrow 14.5\%$) |
| | $\text{Enc}_{\text{public}}$ | $\text{Enc}_{\text{public}}$ | 33.60 | 0 ($\downarrow 100\%$) |
| | $\text{Enc}_{\text{private}}$ | $\text{Enc}_{\text{public}}$ | 31.57 | 54 ($\downarrow 91.8\%$) |
| | $\text{Enc}_{\text{public}}$ | $\text{Enc}_{\text{private}}$ | 22.90 | 451 ($\downarrow 31.3\%$) |
| | $\text{Enc}_{\text{DP}}$ | $\text{Enc}_{\text{DP}}$ | 22.83 | 540 ($\downarrow 16.9\%$) |
| | $\text{Enc}_{\text{private}}$ | $\text{Enc}_{\text{DP}}$ | 21.57 | 603 ($\downarrow 7.2\%$) |
| | $\text{Enc}_{\text{DP}}$ | $\text{Enc}_{\text{private}}$ | 21.62 | 594 ($\downarrow 8.6\%$) |
| | $\text{Enc}_{\text{mixed}}$ | $\text{Enc}_{\text{mixed}}$ | 21.10 | 601 ($\downarrow 8.4\%$) |
| | $\text{Enc}_{\text{private}}$ | $\text{Enc}_{\text{mixed}}$ | 21.12 | 545 ($\downarrow 16.9\%$) |
| | $\text{Enc}_{\text{mixed}}$ | $\text{Enc}_{\text{private}}$ | 21.43 | 498 ($\downarrow 24.1\%$) |

Table 5: Perplexity and data extraction risks for $k$NN-LMs with different encoders for keys ($\text{Enc}_K$) and queries ($\text{Enc}_Q$). Risk reduction (%) compared to the first row is annotated. Privacy measurements are computed using the top 5000 candidates. The $\text{Enc}_{\text{DP}}$ encoder is fine-tuned using DP-SGD with privacy budget $\varepsilon = 10.0$. The $\text{Enc}_{\text{mixed}}$ encoder is fine-tuned using a mix of public and private data points. Results suggest that using different encoders for keys and queries or $\text{Enc}_{\text{mixed}}$ can improve the privacy-utility trade-off. Table 13 in Appendix D presents similar findings on the Medical Transcriptions dataset.

into $k$NN-LMs' datastore, it could potentially aid in reducing privacy risks in applications that prioritize privacy. This becomes particularly relevant in light of previous findings, which suggest substantial privacy leakage from a private datastore.

**Fine-tuning encoders on private data with DP-SGD** Differentially private stochastic gradient descent (DP-SGD) (Abadi et al., 2016) is a recipe for training a deep learning model with differential privacy (Dwork et al., 2006b) guarantee to protect privacy leakage. It operates by modifying the mini-batch stochastic optimization process through the use of per-example gradient clipping and Gaussian noise injection (See Appendix C for details).

**Fine-tuning encoders on a mixture of public and private data** However, adding public data can potentially lead to a decrease in retrieval performance as there is a distribution gap between the public data (e.g., Web Crawl data) used to construct the datastore and the private data (e.g., email conversations) used for encoder fine-tuning. To address this issue, we propose further fine-tuning the encoder on a combination of public and private data to bridge the distribution gap and improve retrieval accuracy. The ratio for combining public and private datasets will be determined empirically through experimentation.

Similarly to Section 5.2, we could also employ separate encoders for keys and queries in the context of untargeted risks, which allows for more precise control over privacy preservation.

### 6.2 Experimental Results

We mainly present our findings using the Enron Email dataset. In Appendix B, we provide results from the Medical Transcriptions dataset, and those findings align with our main findings.

Table 4 demonstrates that when a privately fine-tuned model $\text{Enc}_{\text{private}}$ serves as the encoder, replacing the private datastore $\mathcal{D}_{\text{private}}$ with a public one $\mathcal{D}_{\text{public}}$ in $k$NN-LMs considerably lowers the privacy risk. Furthermore, when using $\text{Enc}_{\text{private}}$ and $\mathcal{D}_{\text{public}}$, the risk level is slightly lower than when using the standard language model with $\text{Enc}_{\text{private}}$ because the model's final response has been interpolated with non-sensitive information, which helps to reduce privacy risks.

Using a public datastore reduces privacy risk but also results in a sudden drop in utility. If more stringent utility requirements but less strict privacy constraints are necessary, adding a few private examples to the public datastore, as shown in Table 4, may also be a suitable solution.

Table 5 demonstrates that using different encoders for keys ($\text{Enc}_K$) and queries ($\text{Enc}_Q$) is more effective in achieving a desirable balance between privacy and utility when using $\mathcal{D}_{\text{public}}$ as the datastore. Specifically, using $\text{Enc}_{\text{private}}$ to encode keys and $\text{Enc}_{\text{public}}$ to encode queries significantly reduces the risk of data extraction with only a slight decrease in perplexity.

We also note that fine-tuning the encoder using DP-SGD only helps slightly reduce the extraction

risk, despite the relatively strict privacy budget $\varepsilon = 10.0$. This is because due to the existence of a private datastore, each inference query in the $k$NN-LM process incurs supplementary privacy costs, leading to the final $k$NN-LM model not satisfying the $\varepsilon$-Differential Privacy criteria.

We further try fine-tuning the encoder using a combination of public and private data, which results in $\text{Enc}_{\text{mixed}}$. The training dataset comprises the entire set of private data of size $N_{priv}$ and $N_{priv} \times r$ public data, where $r$ takes values from $\{0.01, 0.02, 0.05, 0.1, 0.2, 0.5, 1.0\}$. We present attack results using $r = 0.05$ as it achieves the best perplexity. As shown in Table 5, when the encoder is fine-tuned using a combination of public and private data, the perplexity can be enhanced from 21.46 to 21.10 while simultaneously reducing privacy risk. This is because $\text{Enc}_{\text{mixed}}$ helps close the distribution gap between private and public data thus improving the retrieval results. Similarly, using separate $\text{Enc}_K$ and $\text{Enc}_Q$ also helps further reduce the privacy risk.

## 7 Related Work

### 7.1 Retrieval-based Language Models

Retrieval-based language models (Khandelwal et al., 2020; Borgeaud et al., 2022; Izacard et al., 2022; Zhong et al., 2022; Min et al., 2023; Shi et al., 2023) have been widely studied in recent years. These models not only rely on encoder forward running but also leverage a non-parametric component to incorporate more knowledge from an external datastore during inference. The retrieval process starts by using the input as a query, and then retrieving a set of documents (i.e., sequences of tokens) from a corpus. The language model finally incorporates these retrieved documents as additional information to make its final prediction. While the deployment of retrieval-based language models has been shown to lead to improved performance on various NLP tasks, including language modeling and open-domain question answering, it also poses concerns about data privacy.

### 7.2 Privacy Risks in Language Models

Language models have been shown to tend to memorize (Carlini et al., 2019; Thakkar et al., 2021; Zhang et al., 2021; Carlini et al., 2023; Asai et al., 2023) their training data and thus can be prompted to output text sequences from its training data (Carlini et al., 2021), as well as highly sensitive informa-

tion such as personal email addresses (Huang et al., 2022) and protected health information(Lehman et al., 2021; Pan et al., 2020). Recently, the memorization effect in LMs has been further exploited in the federated learning setting (Konečný et al., 2016), where in combination with the information leakage from model updates (Melis et al., 2019; Huang et al., 2020), the attacker is capable of recovering private text in federated learning (Gupta et al., 2022). To mitigate privacy risks, there is a growing interest in making language models privacy-preserving (Yu et al., 2022; Li et al., 2022; Shi et al., 2022b; Yue et al., 2023; Cummings et al., 2023) by training them with a differential privacy guarantee (Dwork et al., 2006b; Abadi et al., 2016) or with various anonymization approaches (Nakamura et al., 2020; Biesner et al.).

Although previous research has demonstrated the potential risks of data extraction in parametric language models, our study is the first investigation of the privacy risks associated with retrieval-based language models; we also propose strategies to mitigate them. The closest effort is Arora et al. (2022), which explores the privacy concerns of using private data in information retrieval systems and provides potential mitigations. However, their work is not specifically tailored to the context of retrieval-based language models.

## 8 Conclusion

This work presents the first study of privacy risks in retrieval-based language models, specifically focusing on $k$NN-LMs. Our objective is to investigate designs and training methodologies for $k$NN-LMs that strike a better privacy-utility trade-off.

There are several conclusions from our investigation. First, our empirical study reveals that incorporating a private datastore in $k$NN-LMs leads to increased privacy risks (both targeted and untargeted) compared to parametric language models trained on private data. Second, for targeted attacks, our experimental study shows that sanitizing $k$NN-LMs to remove private information from both the datastore and encoders, and decoupling the encoders for keys and queries can eliminate the privacy risks without sacrificing utility, achieving perplexity of 16.38 (vs. 16.12). Third, for untargeted attacks, our study shows that using a public datastore and training the encoder on a combination of public and private data can reduce privacy risks at the expense of reduced utility by 24.1%, with perplexity of 21.12 (vs. 16.12).

## Limitations

We discuss the limitations of this work as follows.

- The current study mainly demonstrates the privacy implications of nearest neighbor language models, but there are many other variants of retrieval-based language models, such as RETRO (Borgeaud et al., 2022) and Atlas (Izacard et al., 2022). Further study is needed to understand privacy implications of these models and whether our findings apply.

- In the current study, we use WikiText-103 as the public domain for Enron Email, and PubMed-Patients for Medical Transcriptions. While we believe that these choices of public datasets are realistic, it is important to recognize that this selection may restrict the generalizability of our findings. We acknowledge this limitation and leave the exploration of alternative options for the public dataset as a direction for future work.

- Furthermore, an unexplored aspect of our study is the potential combination of proposed strategies, such as decoupling keys and query encoders, with more diverse privacy-preserving techniques.

## Acknowledgement

This project is supported by an NSF CAREER award (IIS-2239290), a Sloan Research Fellowship, a Meta research grant, and a Princeton SEAS Innovation Grant. We would like to extend our sincere appreciation to Dan Friedman, Jiayi Geng, Zirui Wang, Alexander Wettig, and Zhiyuan Zeng for their valuable comments and feedback on earlier versions of this work.

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

# A Training Data Extraction Attack

## A.1 Untargeted Attack

Carlini et al. (2021) proposes the first attack that can extract training data from a trained language model. The attack consists of two steps: 1) generate candidate reconstructions via prompting the trained models, and 2) sort the generated candidates using a score that implies the possibility of being a memorized text.

**Generate candidate reconstructions** The attacker generates candidates for reconstructions via querying the retrieval-augmented LM's sentence completion API with contexts. Following Carlini et al. (2021)[5], we randomly select chunks from a subset of Common Crawl [6] to feed as these contexts.

**Sort candidates by calibrated perplexity** The second step is to perform membership inference on candidates generated from the previous step. We are using the calibrated perplexity in our study, which has been shown to be the most effective membership metric among all tested ones by Carlini et al. (2021).

The perplexity measures how likely the LM is to generate a piece of text. Concretely, given a language model $f_\theta$ and a sequence of tokens $\mathbf{x} = x_1, \ldots, x_l$, $\mathrm{Perplexity}(f_\theta, \mathbf{x})$ is defined as the exponentiated average negative log-likelihood of $\mathbf{x}$:

$$\exp\left(-\frac{1}{n}\sum_{i=1}^{l} \log f_\theta\left(x_i \mid x_1, \ldots, x_{i-1}\right)\right) \quad (1)$$

A low perplexity implies a high likelihood of the LM generating the text; For a retrieval-augmented LM, this may result from the LM has been trained on the text or has used the text in its datastore.

However, perplexity may not be a reliable indicator for membership: common texts may have very low perplexities even though they may not carry privacy-sensitive information. Previous work (Carlini et al., 2019, 2021) propose to filter out these uninteresting (yet still high-likelihood samples) by comparing to a second LM which never sees the

---

[5]They have empirically show that sampling conditioned on Internet text is the most effective way to identify memorized content, compared with top-$n$ sampling (Fan et al., 2018) and temperature-base sampling (see Section 5.1.1 in their paper).

[6]Common Crawl is a nonprofit organization that crawls the web and freely provides its archives and datasets to the public. See their webpage for details: http://commoncrawl.org/

| Phone | Email | URL |
|---|---|---|
| If you have questions, please feel free to give me a call at | For more information, send email to | The site can be found at |
| Please advise or call me at | For more information please email us at | For more information, visit |
| Please call us at | Suggestions and feedback are welcome at | Please visit our web site at |
| I can be reached at | For more information please email us at | Visit our home page at |
| If you have any questions, please call | Please forward this e-mail to | For more details go to |

Table 6: Example extraction prompts for different types of PIIs.

| MODEL | EVAL. PPL | TARGETED ATTACK | | | | UNTARGETED ATTACK |
|---|---|---|---|---|---|---|
| | | TOTAL | PHONE | EMAIL | URL | # GOOD RECON |
| $k$NN-LM $\text{Enc}_{\text{private}}$ W/ $\mathcal{D}_{\text{private}}$ | 16.12 | 54 | 25 | 23 | 6 | 656 |
| $k$NN-LM, $\text{Enc}_{\text{DP},\varepsilon=5.0}$ W/ $\mathcal{D}_{\text{private}}$ | 29.17 | 7 | 3 | 1 | 3 | 54 |
| $k$NN-LM, $\text{Enc}_{\text{DP},\varepsilon=5.0}$ W/ $\mathcal{D}_{\text{private}}$ | 17.55 | 44 | 21 | 17 | 6 | 651 |

Table 7: Perplexity and data extraction risks for original and DP models on the Enron Email dataset.

private dataset. Specifically, given a piece of text $x$ and the target model $f_\theta$, and the reference LM $f_\theta^{ref}$, the calibrated perpelxity computes the ratio $\text{Perplexity}(f_\theta, \mathbf{x})/\text{Perplexity}(f_\theta^{ref}, \mathbf{x})$.

## A.2 Targeted Attack

The untargeted attack has demonstrated the feasibility of recovering an entire sentence from the deployed retrieval-augmented LM. However, it is possible that only a small segment of a sentence contains sensitive information that can act as personal identifiers, and thus be of interest to the attacker. Therefore, we also consider the type of attack which specifically targets this type of information.

We define personal identifiers and describe the attack method and evaluation subsequently.

### A.2.1 Definition of Personal Identifiers

Personal Identifiable Information (PII) refers to any data that can be used to identify a specific individual, such as date of birth, home address, email address, and telephone number. PII is considered sensitive information and requires proper protection to ensure privacy.

The exact definition of PII can vary depending on the jurisdiction, country, and regulations in place. One of the clearest definitions of PII is provided by Health Insurance Portability and Accountability Act (HIPAA) (Centers for Medicare & Medicaid Services, 1996), which includes name, address, date, telephone number, fax number, email address, social security number, medical record number, health plan beneficiary number, account number, certificate or license number, vehicle

identifiers and serial numbers, web URL, IP Address, finger or voice print, photographic image, and any other characteristic that could uniquely identify the individual. In our study, we focus on three frequently investigated PII in previous literature (Huang et al., 2022; Carlini et al., 2021), including email addresses, telephone numbers, and URLs.

### A.2.2 The Attack

It's important to note that our approach differs from the work of Huang et al. (2022), which aims to reconstruct the relationship between PIIs and their owners[7]. Instead, our study focuses on reconstructing the *actual values* of PIIs. This is because, even if the attacker cannot determine the relationship through the current attack, the reconstruction of PIIs is already considered identity theft[8]. Further, the attacker can use the linkage attack(Narayanan and Shmatikov, 2008) with the aid of publicly available information to determine the relationship between PIIs and their owners.

**PII extraction.** Similar to the training data extraction attack, the PII extraction attack consists of two steps: 1) generate candidate reconstructions, and 2) sort them using membership metrics.

To tailor the attack to recover personal identifiable information rather than the entire training text, we customize the attack prompts based on the type of information to be extracted.

---

[7]This threat model requires additional information about the presence of the owners in the dataset.

[8]https://en.wikipedia.org/wiki/Identity_theft.

| PII | # EMAILS CONTAINING PII | # UNIQUE VALUES |
|---|---|---|
| EMAIL | 1,013 | 758 |
| PHONE | 1,921 | 1,621 |
| URL | 1,641 | 1,396 |

Table 8: Number of PIIs in the Enron Email dataset.

| MODEL | EVAL. PPL |
|---|---|
| (PARAMETRIC LM) $\text{Enc}_{\text{public,WikiText}}$ | 31.42 |
| (PARAMETRIC LM) $\text{Enc}_{\text{public,PubMed}}$ | 23.83 |
| ($k$NN-LM) $\text{Enc}_{\text{public,WikiText}}$ W/ $\mathcal{D}_{\text{private}}$ | 14.75 |
| ($k$NN-LM) $\text{Enc}_{\text{public,PubMed}}$ W/ $\mathcal{D}_{\text{private}}$ | 9.70 |

Table 9: Evaluation perplexity of parametric LMs and $k$NN-LMs with different choices of $\mathcal{D}_{\text{public}}$. Using PubMed as $\mathcal{D}_{\text{public}}$ results in better perplexity.

## B  Experimental Details

**Hyper-parameters**  For each model configuration, we search hyper-parameters $k \in \{64, 128, 256, 512, 1024, 2048\}$ and $\lambda \in \{0.1, 0.2, 0.3, 0.4, 0.5\}$ on a held-out validation set for the best model perplexity.

**PIIs in Enron Email dataset**  We use regular expressions to identify and extract three types of personal identifiers from the Enron Email training dataset for the use of the targeted attack, including telephone numbers, email addresses, and URLs. Table 8 provides statistics for these personal identifiers.

**Prompts for the targeted attack**  We gather common preceding context for telephone numbers, email addresses, and URLs, and use them as prompts for the targeted attack. Table 6 provides example prompts we use in the attack.

**Attack parameters**  For the untargeted attack, we generate 100,000 candidates, and for the targeted attack, we generate 10,000 candidates. We use beam search with repetition penalty = 0.75 for the generation.

## C  Defending the Untargeted Attack with Differential Privacy

### C.1  Differential privacy and DP Stochastic Gradient Descent

*Differential privacy (DP)* (Dwork et al., 2006b,a) is a mathematical framework for ensuring the privacy of individuals in datasets. It can provide a strong guarantee of privacy by allowing data to be analyzed without revealing sensitive information

about any individual in the dataset. Formally, a randomized algorithm $\mathcal{A}$ is $(\varepsilon, \delta)$-*DP* if for any two neighboring datasets $\mathcal{D}$ and $\mathcal{D}'$ (i.e., datasets that differ by a single individual's data), and any subset $\mathcal{S}$ of outputs, it holds that

$$\Pr[\mathcal{A}(\mathcal{D}) \in \mathcal{S}] \leq e^{\varepsilon} \cdot \Pr[\mathcal{A}(\mathcal{D}') \in \mathcal{S}] + \delta.$$

Here, $\varepsilon \in \mathbb{R}_{>0}, \delta \in [0, 1)$ are privacy parameters quantifying the privacy guarantee of the algorithm.

*DP Stochastic Gradient Descent* (DP-SGD) (Abadi et al., 2016) is a recipe for training a deep learning model with DP by modifying the mini-batch stochastic optimization process through the use of per-example gradient clipping and Gaussian noise injection. When training an ML model $f$ parameterized by $\theta$ with the per-example loss function $\ell(\cdot, \cdot)$[9] on dataset $\mathcal{D}$, each optimization step $t$ involves randomly sampling a mini-batch $\mathcal{B}_t$. Given $\mathcal{B}_t$, DP-SGD starts by computing the per-example gradient for each $(x_i, y_i) \in \mathcal{B}_t$, where $x_i$ is the feature vector and $y_i$ is the corresponding label, as follows:

$$\mathbf{g}_t(x_i, y_i) \leftarrow \nabla_{\theta_t} \ell(f_{\theta_t}(x_i), y_i).$$

It then *clips* the gradient $\ell_2$-norm to a maximum $\ell_2$-norm of $C$:

$$[\mathbf{g}_t(x_i, y_i)]_C := \mathbf{g}_t(x_i, y_i) / \max(1, \tfrac{\|\mathbf{g}_t(x_i, y_i)\|_2}{C}).$$

Finally, it produces the private gradient $\hat{\mathbf{g}}_t$ by injecting Gaussian noise into the sum of the clipped per-example gradients:

$$\hat{\mathbf{g}}_t \leftarrow \tfrac{1}{\|\mathcal{B}_t\|} \left( \sum_i [\mathbf{g}_t(x_i, y_i)]_C + \mathcal{N}(0, \sigma^2 C^2 \mathbf{I}) \right),$$

where $(0, \sigma^2 C^2 \mathbf{I})$ is a Gaussian distribution with mean 0 and covariance $\sigma^2 C^2 \mathbf{I}$, and the noise multiplier $\sigma$ is computed from $(\varepsilon, \delta)$ by inverse privacy accounting (e.g., Abadi et al. (2016)).

### C.2  Results

We also evaluate whether DP can mitigate extraction risks in $k$NN-LMs. Specifically, we fine-tune the pre-trained LM on the private dataset with DP-SGD. We vary the privacy budget $\varepsilon$ and fix the failure probability $\delta$ to be $1/N$, where $N$ is the number of training examples. It's important to acknowledge that due to the utilization of a private datastore, each inference query in the $k$NN-LM

---

[9]The specific loss depends on the particular task and model (e.g. cross-entropy loss for classification)

| | |
|---|---|
| **Example #1**: PAST MEDICAL HISTORY:, He has difficulty climbing stairs, difficulty with airline seats, tying shoes, used to public seating, and lifting objects off the floor. He exercises three times a week at home and does cardio. He has difficulty walking two blocks or five flights of stairs. Difficulty with snoring. He has muscle and joint pains including knee pain, back pain, foot and ankle pain, and swelling. He has gastroesophageal reflux disease... | |
| **Example #2**: HISTORY OF PRESENT ILLNESS :,This is a 55-year-old female with a history of stroke, who presents today for followup of frequency and urgency with urge incontinence. This has been progressively worsening, and previously on VESIcare with no improvement. She continues to take Enablex 50 mg and has not noted any improvement of her symptoms. The nursing home did not do a voiding diary. She is accompanied by her power of attorney... | |
| **Example #3**: EXAM: , Ultrasound examination of the scrotum.,REASON FOR EXAM: , Scrotal pain.,FINDINGS: ,Duplex and color flow imaging as well as real time gray-scale imaging of the scrotum and testicles was performed. The left testicle measures 5.1 x 2.8 x 3.0 cm. There is no evidence of intratesticular masses. There is normal Doppler blood flow. The left epididymis has an unremarkable appearance. There is a trace hydrocele... | |
| **Example #4**: TESTICULAR ULTRASOUND,REASON FOR EXAM: ,Left testicular swelling for one day.,FINDINGS: ,The left testicle is normal in size and attenuation, it measures 3.2 x 1.7 x 2.3 cm. The right epididymis measures up to 9 mm. There is a hydrocele on the right side. Normal flow is seen within the testicle and epididymis on the right.,The left testicle is normal in size and attenuation, it measures 3.9 x 2.1 x 2.6 cm... | |
| **Example #5**: PHYSICAL EXAMINATION: , The patient is a 63-year-old executive who was seen by his physician for a company physical. He stated that he was in excellent health and led an active life. His physical examination was normal for a man of his age. Chest x-ray and chemical screening blood work were within normal limits. His PSA was elevated.,IMAGING:,Chest x-ray: Normal.,CT scan of abdomen and pelvis: No abnormalities... | |

Table 10: Examples from the Medical Transcriptions dataset.

| MODEL | EVAL. PPL | # GOOD RECON |
|---|---|---|
| (PARAMETRIC LM) $\text{Enc}_{\text{public}}$ | 23.83 | 0 |
| (PARAMETRIC LM) $\text{Enc}_{\text{private}}$ | 12.00 | 769 |
| ($k$NN-LM) $\text{Enc}_{\text{public}}$ W/ $\mathcal{D}_{\text{private}}$ | 9.70 | 122 |
| ($k$NN-LM) $\text{Enc}_{\text{private}}$ W/ $\mathcal{D}_{\text{private}}$ | 6.61 | 812 |

Table 11: Perplexity and data extraction risks for various model configurations with the medical transcription dataset. The configuration with the highest leakage is emphasized in red ; the configuration with the lowest leakage is highlighted in green . Privacy measurements are computed using the top 5000 candidates for the untargeted attack.

| DATASTORE | | EVAL. PPL | # GOOD RECON |
|---|---|---|---|
| $N_{\text{pub}}$ | $N_{\text{priv}}$ | | |
| 0 | ALL | 6.61 | 812 |
| 0 | 0 | 12.00 | 769 ($\downarrow$ 5.3%) |
| ALL | 0 | 11.32 | 759 ($\downarrow$ 6.5%) |
| | 1,000 | 10.54 | 773 ($\downarrow$ 4.8%) |
| | 2,000 | 9.29 | 787 ($\downarrow$ 3.1%) |
| | 3,000 | 8.26 | 799 ($\downarrow$ 1.6%) |
| | ALL | 6.84 | 804 ($\downarrow$ 1.0%) |

Table 12: Perplexity and data extraction risks for $k$NN-LMs with different numbers of public ($N_{\text{pub}}$) and private ($N_{\text{priv}}$) examples in the datastore. The evaluation uses the medical transcription dataset. The encoder in use is the privately fine-tuned encoder $\text{Enc}_{\text{private}}$. Privacy measurements are computed using the top 5000 candidates.

| $\text{Enc}_{\text{K}}$ | $\text{Enc}_{\text{Q}}$ | EVAL. PPL | # GOOD RECON |
|---|---|---|---|
| $\mathcal{D}_{\text{private}}$ W/ $\text{Enc}_{\text{private}}$ | | 6.61 | 812 |
| $\text{Enc}_{\text{private}}$ | $\text{Enc}_{\text{private}}$ | 11.32 | 759 ($\downarrow$ 6.5%) |
| $\text{Enc}_{\text{public}}$ | $\text{Enc}_{\text{public}}$ | 22.55 | 0 ($\downarrow$ 100%) |
| $\text{Enc}_{\text{private}}$ | $\text{Enc}_{\text{public}}$ | 23.01 | 7 ($\downarrow$ 99.1%) |
| $\text{Enc}_{\text{public}}$ | $\text{Enc}_{\text{private}}$ | 11.36 | 689 ($\downarrow$ 15.1%) |
| $\text{Enc}_{\text{mixed}}$ | $\text{Enc}_{\text{mixed}}$ | 12.32 | 773 ($\downarrow$ 4.8%) |
| $\text{Enc}_{\text{private}}$ | $\text{Enc}_{\text{mixed}}$ | 12.47 | 707 ($\downarrow$ 12.9%) |
| $\text{Enc}_{\text{mixed}}$ | $\text{Enc}_{\text{private}}$ | 12.36 | 692 ($\downarrow$ 14.8%) |

Table 13: Perplexity and data extraction risks for $k$NN-LMs with different encoders for keys ($\text{Enc}_{\text{K}}$) and queries ($\text{Enc}_{\text{Q}}$). The datastore in use is the public datastore $\mathcal{D}_{\text{public}}$. The evaluation uses the medical transcription dataset. Privacy measurements are computed using the top 5000 candidates. The $\text{Enc}_{\text{mixed}}$ encoder is fine-tuned using a mix of public and private data points. Results suggest that using different encoders for keys and queries or $\text{Enc}_{\text{mixed}}$ can potentially improve the privacy-utility trade-off.

process incurs supplementary privacy costs, leading to the final $k$NN-LM model not satisfying the $(\varepsilon, \delta)$-Differential Privacy criteria.

As demonstrated in Table 7, when $\varepsilon$ is set to 5.0, the model showcases minimal utility alongside a marginal privacy risk. Conversely, with $\varepsilon$ raised to 10.0, the utility closely resembles that of utilizing $\text{Enc}_{\text{private}}$ in conjunction $\mathcal{D}_{\text{private}}$, while concurrently slightly reducing the associated risk. These results suggest that DP is also a viable alternative in improving the utility-privacy trade-off in $k$NN-LM.

## D Untargeted Attacks on Medical Transcriptions Dataset

We primarily showcase our findings using the Enron Email dataset in the main paper, as its inclusion of personally identifiable information (PII) enables us to effectively evaluate both targeted and untargeted attacks. To validate our findings, we hereby replicate our experiments specifically for

untargeted attacks on the Medical Transcriptions dataset.

## D.1 Experimental setup

The Medical Transcriptions dataset[10] contains 5,000 medical transcriptions for various medical specialties. We use a subset of 4,500 examples for training and the rest for evaluation. Table 10 provides examples from the dataset. When using the Medical Transcriptions dataset as $\mathcal{D}_{\text{private}}$, we opt for PubMed-Patient[11] as $\mathcal{D}_{\text{public}}$, rather than WikiText. This choice is motivated by the fact that PubMed-Patient exhibits closer semantic alignment with the Medical Transcriptions, leading to enhanced utility in our evaluation. Further insights and justifications regarding this dataset selection can be found in Table 9. We use the GPT-2 base model (Radford et al., 2019) as $\text{Enc}_{\text{public}}$, and fine-tune it on the Medical Transcriptions dataset as $\text{Enc}_{\text{private}}$.

## D.2 Results

The preliminary findings presented in Table 11 align with the observations outlined in Section 4, highlighting a deficiency in the robustness of privacy protection within the design of $k$NN-LMs. Specifically, these results indicate that the privacy concern stemming from the private datastore $\mathcal{D}_{\text{private}}$ outweighs that resulting from the privately fine-tuned model $\text{Enc}_{\text{private}}$.

One approach to address this issue is to incorporate a public datastore (Section 6), which helps mitigate privacy risks - A small number of private examples can be introduced to the public datastore, striking a balance between utility and privacy considerations. Table 12 demonstrates the effectiveness of this approach, offering a promising compromise.

We also observe on the Medicial Transcriptions dataset that separating the key and query encoders yields better results in striking a favorable trade-off between privacy and utility. As shown in Table 5, employing distinct encoders, e.g., $\text{Enc}_{\text{private}}$ for encoding keys and $\text{Enc}_{\text{public}}$ for encoding queries, substantially diminishes the likelihood of data extraction while only marginally affecting perplexity.

---

[10]https://www.kaggle.com/datasets/tboyle10/medicaltranscriptions

[11]PubMed-Patients is derived from 167,000 patient summaries extracted from case reports in PubMed Central. Further details about this dataset can be found at: https://huggingface.co/datasets/zhengyun21/PMC-Patients