# OpenReview forum: "Privacy Implications of Retrieval-Based Language Models"
_EMNLP/2023/Conference — EMNLP 2023 Main_

### Official Review · Reviewer_KGUM · 2023-08-01

**Soundness:** 4

**Excitement:**

3: Ambivalent: It has merits (e.g., it reports state-of-the-art results, the idea is nice), but there are key weaknesses (e.g., it describes incremental work), and it can significantly benefit from another round of revision. However, I won't object to accepting it if my co-reviewers champion it.

**Paper Topic And Main Contributions:**

Large language models have seen tremendous growth more recently. This paper explores privacy implications for a class of large language models called kNN-LMs. As a part of the processing, this class of LMs use a retrieval system to fetch k tokens that are used along with the context in generating the next token. This paper explores how private information that was a part of the training data of this model can leak and suggests ways to mitigate the leakage.

Main contributions of the paper:

• The paper shows how kNN-LMs are susceptible to leakage of private data that could have been a part of the training data of encoders, or available at inference time through a retrieval system.

• The paper shows approaches to mitigate some of the leakage and quantifiably shows the impact. These include sanitizing the training data, using combination of encoders that are training only on public data among others.

**Reasons To Accept:**

This paper delves into a concern around large language models that is becoming increasingly important – that of data leakage, especially when private data is used to train or tune the models. Strengths of the paper are the following:

• For a class of retrieval augmented LLMs, it takes a structured approach at determining leakage of private data for specific kinds of threats.

• It provides a framework that can be used for systematic exploration of privacy leakage in general with LLMs.

• It identifies certain threat models that are realistic representations of what an LLMs might experience and shows the leakage of private data.

• It suggests some approaches to mitigate the leakage, along with the impact it likely has on the performance of the LLM, that way giving the developers of LLMs ways to tradeoff performance and risk of private data leakage.

**Reasons To Reject:**

While the ideas presented in the paper are very pertinent to ongoing research on privacy and LLMs, the major weaknesses of the paper are the following:

• The framework provided would be hard to use with other retrieval augmented LLMs like RETRO, or more recent chain-of-thought based approaches with LLMs.

• The results presented are applicable only to the specific kind of training data used, both public and private. While the results show different ways in which privacy and performance tradeoffs can happen, it is very likely the results would be different for different LLMs, different data for the same kNN-LMs.

• The mitigation strategies – like sanitizing the data – would be very data set specific as well and might not be applicable more generally.

For these reasons, while the framework clearly demonstrates the possibility of data leakage, in its current form, it is not very generalizable and its applicability to other LLMs, other datasets might be limited.

**Reproducibility:**

3: Could reproduce the results with some difficulty. The settings of parameters are underspecified or subjectively determined; the training/evaluation data are not widely available.

**Reviewer Confidence:**

3: Pretty sure, but there's a chance I missed something. Although I have a good feel for this area in general, I did not carefully check the paper's details, e.g., the math, experimental design, or novelty.

**Typos Grammar Style And Presentation Improvements:**

* It would be great if the authors can also explain what w_i is in the Inference part of section 2.1

* Section 5.1 - Fine-tuning encoders on a mixture of public and private data (page 7) - The first sentence "However, adding public data may cause ..." needs to be rephrased.

---

> ### Author Rebuttal · Authors · 2023-08-29
>
> We appreciate the reviewer's thoughtful feedback. It's encouraging to see the reviewer recognizing the significance of our study, the well-structured approach we proposed, and the systematic evaluation we conducted. We are now taking the opportunity to carefully address the detailed comments provided by the reviewer, as this will aid us in revising the paper.
>
>
> **“The framework provided would be hard to use with other retrieval augmented LLMs like RETRO, or more recent chain-of-thought based approaches with LLMs.”**
>
> We acknowledge this is a limitation, but we believe that our work paves the first step toward understanding the trade-off between privacy and utility in retrieval-based LMs. Notably, we observe that $k$NN-LMs are more prone to privacy breaches from their private datastore compared to parametric models. We further explore mitigations of privacy risks, including sanitization of the private datastore and decoupling key and query encoders.
>
> We plan to leave the exploration of other retrieval-based LMs to future work. We also would like to note that $k$NN-LMs have received wide attention in the last few years, and many works have been proposed to improve their efficiency, their training methods, or their downstream adaptations (He et al., 2021; Zhong et al., 2022; Shi et al., 2022a; Xu et al., 2023, in our submission). Hence, we believe that understanding the privacy risks of $k$NN-LMs itself is important, and the scope would suffice in this work.
>
> We are confused by why chain-of-thought is discussed here, as CoT is more of a prompting technique and hasn’t been really investigated in retrieval-augmented LMs (to the best of our knowledge).
>
>
> **“The results presented are applicable only to the specific kind of training data used, both public and private.”**
>
> We respectfully disagree with this point. We would like to note that we also evaluated our approach with the Medical Transcriptions dataset as $D_\textrm{private}$, as shown in Appendix C. We also varied the choice of public dataset $D_\textrm{public}$ among WikiText and PubMed among WikiText and PubMed (see Appendix C.1, Table 7), and found that PubMed was a more suitable public dataset.
>
> The results in Table 8, 10, and 11 for the Medical Transcriptions dataset are consistent with our findings in Table 1, 3, and 4 for the Enron Email dataset. This suggests that our approach is generalizable to different types of private datasets.
>
> **“The mitigation strategies – like sanitizing the data – would be very data set specific.”**
>
> We respectfully disagree with this comment. Given a privacy-sensitive application, as long as we can define “privacy leakage” at the verbatim level, we can operate sanitization.
>
> We further note that the verbatim level definition of “privacy leakage” is general and widely adopted. Notably, regulations such as HIPAA and CCPA offer explicit definitions of privacy-sensitive data. Consequently, these regulatory frameworks can serve as the basis for establishing the verbatim-level definition of "privacy leakage." For example, HIPAA defines 18 identifiers that are considered personally identifiable information (PII), including names, addresses, phone numbers, etc.
>
> Therefore, the strategy of sanitization, despite its prerequisite for a verbatim-level privacy leakage definition, remains applicable on a broader scale.
>
>
> **“What w_i is in the Inference part of section 2.1”**
>
> {$c_i, w_i$} represents an entry in the kNN-LM’s datastore, where $c_i$ is the leftward context and $w_i$ is the next word for $c_i$. Thanks for your suggestion, and we will make it clear in the revision.
>
> **Rephrasing a sentence**
>
> Thanks for your suggestion, and we will fix it in the revision.

---

### Official Review · Reviewer_EAGn · 2023-08-05

**Soundness:** 4

**Excitement:**

4: Strong: This paper deepens the understanding of some phenomenon or lowers the barriers to an existing research direction.

**Paper Topic And Main Contributions:**

In this paper, the authors investigate the privacy risks associated with retrieval-based language models. The main objective of the paper is to find ways to improve the balance between privacy and utility of these models. The authors finds that Including a private datastore in kNN-LMs increases both targeted and untargeted privacy risks compared to parametric LLMS trained on private data. For targeted attacks, sanitizing private information can eliminate privacy risks without sacrificing utility. For untargeted attacks, using a public datastore and training the encoder on a combination of public and private data can reduce privacy risks but leads to a decrease in utility.

**Reasons To Accept:**

This is a well written paper. The topic is interesting and important. The authors motivate the paper well and provide a solid background and problem definition. The experiments are comprehensive and the insights from the results are interesting.

**Reasons To Reject:**

The use of perplexity as a measure of utility is not well motivated.

The nearest neighbor variation of retrieval based language models is one of many different types of retrieval based language models. It would be useful reflect privacy implications in other types of retrieval based language models.



**Reproducibility:**

3: Could reproduce the results with some difficulty. The settings of parameters are underspecified or subjectively determined; the training/evaluation data are not widely available.

**Reviewer Confidence:**

1: Not my area, or paper was hard for me to understand. My evaluation is just an educated guess.

---

> ### Author Rebuttal · Authors · 2023-08-29
>
> We appreciate the reviewer's thoughtful feedback. It's motivating to see the reviewer’s recognition of our study's significance, solid formulation, comprehensive empirical evaluation, and interesting insights. We are now taking the opportunity to carefully address the detailed comments provided by the reviewer, as this will aid us in revising the paper.
>
>
> **"The use of perplexity as a measure of utility is not well motivated."**
>
> We appreciate the question. To clarify, in our context, "utility" refers to the performance of the language model. We use perplexity as a measure of utility following previous works (Khandelwal et al., 2020; He et al., 2021; Zhong et al., 2022; Shi et al., 2022a; Xu et al., 2023, as referenced in our submission), and we will leave the impact on downstream tasks as future work.
>
>
>
> **“It would be useful reflect privacy implications in other types of retrieval based language models”**
>
> We acknowledge this is a limitation, but we believe that our work paves the first step toward understanding the trade-off between privacy and utility in retrieval-based LMs. Notably, we observe that $k$NN-LMs are more prone to privacy breaches from their private datastore compared to parametric models. We further explore mitigations of privacy risks, including sanitization of the private datastore, and decoupling key and query encoders.
>
> We plan to leave the exploration of other retrieval-based LMs to future work. We also would like to note that $k$NN-LMs have received wide attention in the last few years, and many works have been proposed to improve their efficiency, their training methods, or their downstream adaptations (He et al., 2021; Zhong et al., 2022; Shi et al., 2022a; Xu et al., 2023, in our submission). Hence, we believe that understanding the privacy risks of kNN-LMs itself is important, and the scope would suffice in this work.

---

### Official Review · Reviewer_nKXz · 2023-08-05

**Soundness:** 3

**Excitement:**

3: Ambivalent: It has merits (e.g., it reports state-of-the-art results, the idea is nice), but there are key weaknesses (e.g., it describes incremental work), and it can significantly benefit from another round of revision. However, I won't object to accepting it if my co-reviewers champion it.

**Paper Topic And Main Contributions:**

The paper studies the privacy risk of retrieval-based language models (LMs) by designing data extraction attacks. In particular, the authors propose two types of attacks: (1) targeted attack to reconstruct certain segments of text and (2) untargeted attack to recover the entire training example.

**Questions For The Authors:**

1. Could you please provide a justification for the above weaknesses?
2. In Tables 1 and 2, what is the evaluation dataset in terms of eval PPL, $\mathcal{D}_{\text{public}}$ or $\mathcal{D}_{\text{private}}$?
3. In untargeted attacks, the goal is to reconstruct the entire training example. Can you showcase what types of training examples are considered private in your case?
4. The description of nearest neighbor language models is unclear (e.g., notation unclear). For example, $w_i$ is not defined throughout the paper.

**Reasons To Accept:**

1. The paper is well-written and well-motivated. The privacy risks of retrieval-based LMs are important.

2. Experimental results demonstrate that the proposed attacks are effective and the authors provide countermeasures to mitigate the privacy risk.

**Reasons To Reject:**

1. The privacy definition (e.g., memorization) is not formally defined. Carlini et al. (2021) provide a rigorous privacy definition (e.g., k-eidetic memorization). However, the authors do not provide such a definition. Instead, they only use whether to extract segments or a whole training example to differentiate targeted and untargeted attacks. The categorization is strange to me. Besides, in the evaluation of untargeted attacks, the author uses ROUGE-L as the privacy metric, and the choice of privacy metrics does not make sense to me.

2. The threat model is unclear.  For example, in the privacy measurements, the author implicitly assumes the attacker could get access to the private dataset $p_i$; otherwise, there is no way for the attacker to evaluate the top-n candidate. However, this does not make sense to me. The authors should provide more justification for this.

3. The authors do not provide insights into *why* the privacy risk of kNN-LM is more serious than parametric LM empirically or theoretically.

4. The paper misses differential privacy (DP) as a defense in the experiments. Though the authors acknowledge this point in the limitation section, I still think it should be included in the experiment since DP is a standard defense against privacy attacks.

**Reproducibility:**

4: Could mostly reproduce the results, but there may be some variation because of sample variance or minor variations in their interpretation of the protocol or method.

**Reviewer Confidence:**

4: Quite sure. I tried to check the important points carefully. It's unlikely, though conceivable, that I missed something that should affect my ratings.

**Typos Grammar Style And Presentation Improvements:**

In Table 2, the caption indicates the best utility under sanitization is highlighted in boldface. However, I do not see any number highlighted in boldface. Is it a typo?

---

> ### Author Rebuttal · Authors · 2023-08-29
>
> We appreciate the reviewer's thoughtful feedback. It's encouraging that the reviewer acknowledges the importance of our study on privacy risks in retrieval-based LMs and finds our mitigation strategies effective. We will carefully address your comments below, and have added the DP experiments in the rebuttal additionally. We are more than willing to engage in further discussion should any follow-up questions arise!
>
> **“The privacy definition (e.g., memorization) is not formally defined. ”**
>
> We hereby clarify our privacy definition: Our definition of privacy leakage extends the standard definition of the Model Knowledge Extraction (Definition 1 in Carlini et al., 2021)). Specifically, a string $s$ is extractable from an LM $f_θ$ if there exists a prefix $c$ such that: $s \gets \arg\max_{s’} f_\theta(s’|c)$. Namely, the model generates $s$ as the most likely continuation when prompted with some prefix $c$.
>
> Specifically, in our untargeted setting, $s$ is an example in the training dataset, and in the targeted setting, $s$ is a subsequence of a training example. We will provide this definition in the revision.
>
> **"The author uses ROUGE-L as the privacy metric, and the choice of privacy metrics does not make sense to me.”**
>
> ROUGE-L measures the longest common subsequence (LCS) between the attack result and the ground truth, thus representing the fidelity of the attacker results. ROUGE-L score has been used to measure the reconstruction fidelity in previous work [1,2,3]. If the reviewer can specify why it does not make sense, we are happy to further clarify it.
>
> **“The threat model is unclear”**
>
> We believe this is due to a misunderstanding, and we do not assume the attacker has access to the private dataset. The use of private datasets is only for automatic evaluation purposes.
>
> Specifically, our threat model (defined in Section 3.1) assumes that the attacker only has black-box API access to the model: The attacker generates top-n candidates through the model's API and sorts them based on their defined score, all **without** utilizing private data. We then evaluate the attack’s performance by comparing top-n candidates against the private dataset, which aligns with standard practices in reconstruction attacks [1, 2, 3, 4].
>
> Note that our threat model is indeed the same as that by Carlini et al. 2021 in our submission, but they only considered small-scale human evaluation while we used private data for automatic evaluation.
>
> **“Explanation for why kNN-LM is more vulnerable than parametric LM”**
>
> We would like to clarify that we didn’t claim that the privacy risk of $k$NN-LMs is more serious than parametric LMs. Instead, our work focuses on understanding the conditions under which $k$NN-LM exhibits higher or lower vulnerability.
>
> Two notable observations we made are:
> - When private data is incorporated into the datastore, the vulnerability of $k$NN-LM tends to increase, as indicated in Table 1. Our rationale behind this finding is that the datastore offers a more explicit way for potential privacy breaches compared to the parameters within traditional LMs. This inference gains support from our results in Figure 2, where we empirically illustrate that with a larger $\lambda$ (resulting in increased emphasis on the $p_{kNN}$ term during final prediction), the model's vulnerability increases, and with a smaller $k$ (resulting in the model being influenced by a limited group of private nearest neighbors.) the model's vulnerability increases.
> - Conversely, incorporating public data into the datastore along with privately fine-tuned encoders renders $k$NN-LMs less vulnerable than parametric LMs. This reinforces the rationale that the private datastore primarily contributes to the leakage.
>
> We believe that a theoretical understanding of this phoneme would be an interesting area of future research, but it may exceed the scope of current empirical investigation.
>
> **“Misses differential privacy (DP) as a defense”**
>
> We appreciate the suggestion and provide the DP results below. Specifically, we fine-tune the pre-trained LM on the private dataset with DP-SGD. We vary the privacy budget $\epsilon \in \{5.0, 10.0\}$ and fix the failure probability $\delta$ to be $1/N$, where $N$ is the number of training examples. It's important to acknowledge that due to the utilization of a private datastore, each inference query in the $k$NN-LM process incurs supplementary privacy costs, leading to the final $k$NN-LM model **not** satisfying the $(\epsilon, \delta)$-Differential Privacy criteria.
>
> As demonstrated, when $\epsilon$ is set to 5.0, the model showcases minimal utility alongside a marginal privacy risk. Conversely, with $\epsilon$ raised to 10.0, the utility closely resembles that of utilizing Enc$\_\textrm{private}$ in conjunction $D\_\textrm{private}$, while concurrently slightly reducing the associated risk. These results suggest that DP is also a viable alternative in improving the utility-privacy trade-off in $k$NN-LM.
>
> | Model | Eval PPL | Targeted attack, total | Targeted attack, phone | Targeted attack, email | Targeted attack, URL | Untargeted attack (# Good Recon) |
> |---|---|---|---|---|---|---|
> | $k$NN-LM, Enc$_\textrm{private}$  w/ $D\_\textrm{private}$| 16.12 | 54 | 25 | 23 | 6 | 656 |
> | $k$NN-LM, Enc$_\textrm{DP, $\epsilon$=5.0}$  w/ $D\_\textrm{private}$| 29.17 | 7 | 3 | 1 | 3 | 54 |
> | $k$NN-LM, Enc$_\textrm{DP, $\epsilon$=10.0}$  w/ $D\_\textrm{private}$ | 17.55 | 44 | 21 | 17 | 6 | 651 |
>
> We also would like to highlight that DP and our suggested defense mechanisms are not mutually exclusive. We now provide additional evidence of the compatibility between DP and our proposed defense strategies: by disentangling the key and query encoders (with one of the encoders being the DP encoder), we effectively enhance the trade-off between privacy and utility.
>
> | Enc$_K$ | Enc$_Q$ | Eval PPL | Good Recon |
> |---|---|---|---|
> | Enc$_\textrm{private}$ | Enc$_\textrm{private}$ | 21.46 | 561 |
> | Enc$_\textrm{DP, $\epsilon$=10.0}$ | Enc$_\textrm{private}$ | 21.62 | 594 |
> | Enc$_\textrm{private}$ | Enc$_\textrm{DP, $\epsilon$=10.0}$ | 21.57 | 603 |
> | Enc$_\textrm{DP, $\epsilon$=10.0}$ | Enc$_\textrm{DP, $\epsilon$=10.0}$ | 22.83 | 540 |
>
>
> **In Tables 1 and 2, what is the evaluation dataset in terms of eval PPL, $D_{\text{public}}, D_{\text{private}}$?**
>
> Table 1 and 2 uses the evaluation set of Enron Email dataset ($D_{\text{private}}$) for its evaluation.
>
> **Can you showcase what types of training examples are considered private in your case?**
>
> Great question! When it comes to an untargeted attack, any effort that effectively reconstructs a significant portion of the original training example is deemed a successful attack, regardless of the type of training example. This is because the attack only needs to function in the worst-case scenario rather than the average scenario.
>
> For the reviewer's reference, we've also included a few examples from the Enron Email dataset. (We've anonymized any identifiable information to protect against potential privacy breaches inherent in the dataset.)
>
>
> | Example |
> |---|
> | Request ID : 0000000000***** Request Create Date : */*/* 8:27:00 AM Requested For : ***.***@enron.com Resource Name : Market Data Bloomberg Resource Type : Applications |
> | You can reach me over the weekend and in the evening at either ***-***-**** or ***-***-****. |
> | This would give you a total loan of ****, total cost of **** for equity required of ****. |
> | Winter 2001 baseload traded as high as ** pounds a megawatt-hour and as low as ** pounds a megawatt-hour, before closing at ** pounds a megawatt-hour, ** pence lower than Friday. |
> | everything is correct except for password: ***. |
>
> **The description of nearest neighbor language models is unclear (e.g., notation unclear). For example, w_i is not defined throughout the paper.**
>
> {$c_i, w_i$} represents an entry in the $k$NN-LM’s datastore, where $c_i$ is the leftward context and $w_i$ is the next word for $c_i$. Thanks for your suggestion, and we will make it clear in the revision.
>
> **Typo in Table 2**
>
> Thanks for the reviewer’s careful reading. We apologize for the typo and will fix it in revision.
>
> **References**
>
> [1] Deng et al., Tag: Gradient Attack on Transformer-based Language Models. EMNLP 21
>
> [2] Balunovic et al., Lamp: Extracting Text from Gradients with Language Model Priors. NeurIPS 22
>
> [3] Gupta et al., Recovering Private Text in Federated Learning of Language Models. NeurIPS 22
>
> [4] Huang et al., Are Large Pre-Trained Language Models Leaking Your Personal Information? EMNLP 22

---

### Meta-Review · Area_Chair_jDwn · 2023-09-19

**Recommendation:** 4

**Metareview:**

The manuscript studies how to balance training of retrieval-based Language Models between utility (represented by perplexity) and privacy tradeoff. The comprehensive study reveals that kNN-LMs are more susceptible to leaking private information than parametric model as more private data and fewer public data are incorporated into the datastore. The authors propose how to quantitatively mitigate the leakage issue, either by sanitizing the training data or by mixing public and private data. All reviewers agree that this paper provides valuable framework that can systematically explore privacy leakage for a class of retrieval-augmented LLMs and mitigating solutions. To further improve the study, it is encouraged to navigate other types of retrieval-augmented language models and other types of utility, especially multi-task solving capabilities.

---

### Decision · Program_Chairs · 2023-10-07

**Decision:**

Accept-Main

**Comment:**

The manuscript studies how to balance training of retrieval-based Language Models between utility (represented by perplexity) and privacy tradeoff. The comprehensive study reveals that kNN-LMs are more susceptible to leaking private information than parametric model as more private data and fewer public data are incorporated into the datastore. The authors propose how to quantitatively mitigate the leakage issue, either by sanitizing the training data or by mixing public and private data. All reviewers agree that this paper provides valuable framework that can systematically explore privacy leakage for a class of retrieval-augmented LLMs and mitigating solutions. To further improve the study, it is encouraged to navigate other types of retrieval-augmented language models and other types of utility, especially multi-task solving capabilities.